# Advancing Medicine with Lipid-Based Nanosystems—The Successful Case of Liposomes

**DOI:** 10.3390/biomedicines11020435

**Published:** 2023-02-02

**Authors:** Hugo Luiz, Jacinta Oliveira Pinho, Maria Manuela Gaspar

**Affiliations:** Research Institute for Medicines (iMed.ULisboa), Faculty of Pharmacy, Universidade de Lisboa, 1649-003 Lisboa, Portugal

**Keywords:** nanomedicine, liposomes, therapeutic effect, clinical trials

## Abstract

Nanomedicine, a promising area of medicine, employs nanosized tools for the diagnosis, prevention, and treatment of disease. Particularly, liposomes, lipid-based nanovesicles, are currently one of the most successful nanosystems, with extensive applications in the clinic and an increasing pipeline of products in preclinical and clinical development. These versatile nanotechnological tools are biocompatible and biodegradable, and can load a variety of molecules and, ultimately, improve the therapeutic performance of drugs while minimizing undesired side effects. In this review, we provide a brief description on liposomes’ composition and classification and mainly focus on their clinical use in various areas, including disease management (e.g., cancer, fungal and bacterial infections, ocular pathologies), analgesia, vaccination, diagnostics, and immunosuppression in organ transplantation. Herein are described examples of current liposomal products already in the clinic, as well as the most recent clinical trials involving liposomes as effective and safe nanomedicine tools.

## 1. Introduction

The constant evolution of science is prompted by knowledge exchange between different areas. In the biology field, it was in the late 1600s that Robert Hooke carried out the first observations of the unit of life, the cell, with only a thirty-times magnification microscope. In parallel, Anton van Leeuwenhoek developed and improved the microscopy field, constructing a microscope capable of up to 300 times magnification that allowed the observation of different types of mammalian cells, tissues and bacteria [1]. In 1931, Ernst Ruska and Max Knoll, two German scientists, achieved a major breakthrough in microscopy technology by creating the first transmission electron microscope [2].

### 1.1. A Brief History of Liposomes

Since the observation of cells under a microscope, scientists have tried to understand how lipids and biological membranes behave [3]. In 1890, Lord Raleigh studied the interfacial tension between a triglyceride (castor oil) and water [4]. Later, in 1925, Gorter and Grendel demonstrated that the cell membrane was constituted by phospholipid molecules, the “lipid bilayers” [5]. Following these discoveries, Singer and Nicolson suggested the “Fluid Mosaic Model”, in 1972, still accepted today [6]. The introduction of electron microscopy allowed the visualization of biological membranes in greater detail. These appeared as two “opaque” bands divided by “a less opaque interspace” [7], an observation that was interpreted as two opposed phospholipid monolayers [8]. It was around 1962 that Alec Bangham, using a friend’s electron microscope, observed that, in aqueous negative stain, the phospholipid lecithin or its mixture with cholesterol spontaneously formed closed structures, with concentric lamellae [9]. This apparently simple discovery was a revolutionary step in the course of lipid research [3]. At that time, the systems visualized by Bangham were designated as “multilamellar smectic mesophases” and, later on, Gerald Weissmann proposed the name “liposomes” [10]. In the 70 s, Bangham postulated that “something like liposomes must have been available to house the first forms of cellular life” that, together with other studies, remarkably impacted evolutionary history [10].

### 1.2. Liposome Properties and Composition

Liposomes are defined as small synthetic vesicles composed of one or more lipid bilayers separated by aqueous compartments (Figure 1a) [3]. Liposomes are mainly composed of phospholipids, a group of amphiphile molecules that includes two main categories: glycerophospholipids and sphingomyelins [3,11]. Examples of glycerophospholipids are phosphatidyl choline (PC), phosphatidyl ethanolamine (PE), and phosphatidyl glycerol (PG) [12,13]. An important characteristic of lipids that affects the bilayer properties, including its fluidity, is the phase transition temperature (Tc). This parameter is defined as the temperature at which the physical state of the lipid changes from an ordered and rigid gel-state to a disordered and more fluid liquid-crystalline phase [14]. Tc highly depends on the length and saturation degree of nonpolar chains, with longer and more saturated chains corresponding to higher Tc [14]. Examples of phospholipids with different chain length and saturation degree are distearoyl phosphatidyl choline (DSPC; Tc ≈ +55 °C), dipalmitoyl phosphatidyl choline (DPPC; Tc ≈ +41 °C), dimyristoyl phosphatidyl choline (DMPC; Tc ≈ +24 °C), and dioleoyl phosphatidyl choline (DOPC; Tc ≈ −17 °C) (Figure 1b) [13,14].

These lipid vesicles are extremely versatile, as they can load a variety of molecules, protecting them from premature degradation, changing the pharmacokinetics and improving the biodistribution profile, and ultimately enhancing the therapeutic effect of incorporated drugs [15,16]. Different factors can directly influence the properties of developed liposomal formulations, such as lipid composition, surface charge, bilayer fluidity, size, and preparation method [11,13]. For instance, surface modification with polyethylene glycol (PEG) covalently linked to distearoyl phosphatidyl ethanolamine (DSPE-PEG; Figure 1) is able to decrease the opsonization by plasmatic proteins, avoiding premature detection and uptake by the mononuclear phagocytic system [17,18,19]. Consequently, in vivo, this increases the half-life of liposomes in the bloodstream and their ability to extravasate to affected sites [17,18,19].

**Figure 1 biomedicines-11-00435-f001:**
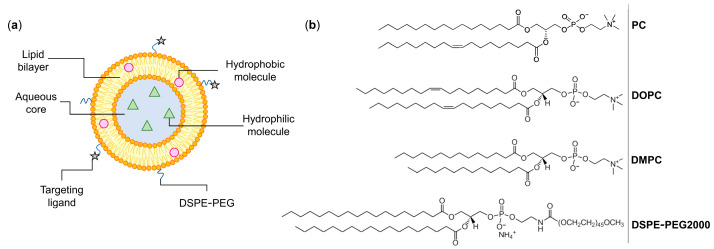
(**a**) Schematic representation of a liposome. These lipid-based nanosystems mimic biological membranes and are composed of one (unilamellar) or more (multilamellar) concentric lipid bilayers separated by aqueous compartments. Liposomes are able to accommodate both hydrophilic and hydrophobic molecules and their surface may be coated with specific ligands that recognize receptors overexpressed at tumor cells. (**b**) Chemical structures of commonly used phospholipids for the preparation of liposomes. PC: phosphatidyl choline; DOPC: dioleoyl phosphatidyl choline; DMPC: dimyristoyl phosphatidyl choline; DSPE-PEG: poly(ethylene glycol) 2000 covalently linked to distearoyl phosphatidyl ethanolamine. Images adapted from [20].

### 1.3. Classification and Main Applications of Liposomes

As depicted in Figure 2, liposomes are usually classified into three big groups based on size and number of bilayers: unilamellar vesicles (ULVs; one lipid bilayer); oligolamellar vesicles (OLVs; 2–5 lipid bilayers); and multilamellar vesicles (MLVs; more than five lipid bilayers, MLVs). In terms of diameter, ULVs are categorized into small unilamellar vesicles (SUVs; 20–100 nm), large unilamellar vesicles (LUVs; 100–1000 nm), and giant unilamellar vesicles (GUVs; >1000 nm) [11,21,22,23]. In some cases, concentric phospholipid spheres are produced within larger liposomes, forming multivesicular vesicles (MVV) [21,24,25,26].

Liposomes can also be further subdivided according to their composition and application (Table 1) [22,26,27].

The first research studies using liposomes began in the the 1960s, being applied as models of biological membranes [3]. In 1971, Gregory Gregoriadis introduced liposomes as delivery systems for enzymes (lysozyme) [28,29] and, since then, this lipid-based nanosystem has been used to entrap L-asparaginase [30,31], catalase [32], superoxide dismutase [33], among other enzymes [34]. Besides functioning as models of biological membranes, liposomes are versatile and ideal nanotechnological tools for various purposes, as detailed in Table 2. 

As previously mentioned, liposomes are extremely useful as delivery systems, with countless examples of high and low molecular weight molecules (e.g., enzymes, proteins, metal-based complexes, antibiotics) that can be loaded in this lipid-based nanoplatform [30,31,33,42,43,44,45,46,47,48,49,50,51,52,53,54,55]. As part of the drug development pipeline, these liposomal formulations must first undergo extensive characterization and evaluation in both in vitro and in vivo models. In Table 3 are presented some examples of preclinical reports with different liposomal formulations, highlighting the advantages of using this nanosystem for therapeutic applications.

## 2. Liposomes as Nanomedicine Tools

Over the years, liposomes have been employed as tools to maximize the therapeutic index of a panoply of molecules, including anticancer drugs, antibiotics, genetic material and antifungals [16,19,27,56,57,58]. Nowadays, the success of liposomes is evidenced by several approved products (Table 4; Figure 3) or undergoing clinical trials [59,60,61].

### 2.1. Liposomes for the Treatment of Fungal Infections

AmBisome^®^ was the first approved liposomal formulation to be used in the clinic against fungal infections, including aspergillosis, mucormycosis, invasive candidiasis, and cryptococcal meningitis [63]. Initially, the drug amphotericin B was developed for the treatment of local mycotic infections, being subsequently approved as a systemic antifungal agent [64]. Due to associated nephrotoxicity and infusion-related reactions, the liposomal formulation AmBisome^®^ was designed. This retained the antifungal activity of the drug, while significantly reducing toxicity [63]. In 2022, results from a phase III clinical trial in patients with HIV-associated cryptococcal meningitis demonstrated that a single dose of AmBisome^®^ (together with flucytosine and fluconazole) was equivalent to the current standard of care, with less toxicity [65].

### 2.2. Liposomes for Cancer Management

In cancer treatment, one of the most well-known examples is Doxil^®^/Caelyx^®^, the commercial name for liposomal doxorubicin. This drug, when intravenously administered in the free form, presents high cardiotoxicity. However, following its encapsulation in liposomes, a drastic toxicity reduction was achieved, while the antitumor efficacy was maintained [66], reinforcing the advantages of using this delivery nanosystem. The loading of daunorubicin into liposomes (DaunoXome^®^) also proved to be very advantageous, increasing tumor drug delivery by approximately 10-fold compared to the free drug and promoting a sustained release in vivo [67].

Another example of an antineoplastic liposomal product is Vyxeos^®^, which was approved for the treatment of acute myeloid leukemia [68]. This nanoformulation contains two cytotoxic drugs, cytarabine and daunorubicin (5:1). As each one of these drugs displays distinct mechanisms of action, the simultaneous delivery of both drugs in a liposomal formulation resulted in a synergistic effect, increasing treatment efficacy with a lower dosing [68,69].

Continuous advances in nanomedicine are witnessed every day, with novel liposomes being developed and entering clinical trials. Liposomes with ligands attached to their surface (immunoliposomes) are currently under investigation to selectively recognize receptors overexpressed at tumor cells and to promote cellular internalization [70]. This, in turn, results in increased therapeutic efficacy and reduced unwanted side effects. For instance, the endothelial growth factor receptor (EGFR) is known to promote tumorigenesis and it is recognized as a biomarker of drug resistance [71]. Anti-EGFR immunoliposomes loading doxorubicin are in phase II clinical trials for patients with advanced triple-negative breast cancer (NCT02833766) [72]. Also, constant improvements of existing nanomedicines are being made. For example, a novel liposomal formulation of doxorubicin (Talidox) was developed and is currently under a phase I clinical trial (NCT03387917) to assess safety, maximal tolerated dose, pharmacokinetics, and preliminary efficacy. This new liposomal product is expected to improve the benefit/risk profile when compared to established doxorubicin liposomes, namely Doxil^®^/Caelyx^®^ [72]. Topotecan, a hydrophilic anticancer drug derived from camptothecin, was encapsulated in dihydrosphingomyelin-based liposomes (FF-10850). Currently, liposomal topotecan is undergoing phase I clinical trials (NCT04047251) for the treatment of advanced solid tumors [72,73]. Cancer resistance to drug therapy is a challenge that greatly affects clinical outcomes. In the case of platinum-resistant small cell lung cancer, Onivyde^®^ (liposomal irinotecan) is being tested in a phase 3 clinical trial, showing promising antitumor activity and safety [74].

### 2.3. Liposomes for the Delivery of Antibacterial Drugs

Liposomes also function as tools to enhance the therapeutic performance of antibiotics, being advantageous against antimicrobial resistance [56]. Arikayce^®^ is a liposomal formulation of amikacin, an antibiotic that belongs to the aminoglycoside class [75,76]. Due to limited safety data, the Food and Drug Administration (FDA) approved this liposomal product only for adult patients with nontuberculous mycobacterial lung disease, in a combination treatment regimen. Inhalation of Arikayce^®^ through a nebulizer improves lung drug delivery compared to intravenously administered free amikacin, effectively clearing pulmonary infections caused by *Mycobacterium avium* complex [75,76]. A clinical trial (NCT04163601) with liposomal amikacin for inhalation is being conducted to assess the therapeutic efficacy against infections caused by *Mycobacterium abscessus*, which are difficult to treat and are commonly found in patients with cystic fibrosis [72].

### 2.4. Liposomes for Ophthalmologic Applications

In ocular diseases, Visudyne^®^ is used for the therapeutic management of age-related macular degeneration by photodynamic therapy [77]. The treatment starts with an intravenous infusion of Visudyne^®^, followed by nonthermal light activation of the photosensitive drug, verteporfin [77]. Moreover, for the relief of dry eye symptoms, the liposomal products Lacrisek^®^ (Fidia Farmaceutici S.p.A., Abano Terme, Italy) and Optrex™ ActiMist™ (Reckitt Bensicker, Slough, UK) are currently commercialized [78]. In the prevention of macular edema after femtosecond laser-assisted cataract surgery, liposomes loading triamcinolone acetonide were as effective as a combination therapy, with better visual outcomes [79]. Furthermore, in a phase I/II clinical trial (NCT02006147), a liposome ophthalmic formulation of dexamethasone sodium phosphate (TLC399) effectively improved macular edema secondary to retinal vein occlusion [72,80].

### 2.5. Liposomes in Analgesia

Another useful application of liposomes is in pain management. Opioid analgesia is a major part of post-operative pain control. However, the excessive prescription of opioids brings troubling health consequences for the patients. DepoDur™ (Pacira Pharmaceuticals, Inc., San Diego, CA, USA) is a liposomal formulation of morphine sulfate for a single-dose administration into the lumbar epidural space, providing up to 48 h of analgesia [81,82]. Compared to standard epidural morphine, this nanoformulation decreased the need for supplemental analgesics and ameliorated post-Cesarean delivery pain, with improved mobility of patients [81,82]. In addition, liposomal bupivacaine (Exparel^®^) is a long-acting analgesia formulation that effectively decreased post-surgery pain and reduced opioid needs [83]. Furthermore, the analgesic efficacy of a single dose of liposomal bupivacaine is being assessed in a phase IV study (NCT03737604) in renal transplant recipients [72].

### 2.6. Liposomes in Vaccination

In the field of immunization, the introduction of mRNA vaccines was a breakthrough since these elicit a potent and long-lasting immunity. The application of lipid-based nanosystems for mRNA vaccination constitutes an effective strategy and, with the 2019 outbreak of severe acute respiratory syndrome coronavirus 2 (SARS-CoV-2), this nanotechnology further proved its importance in the response against this global health crisis [84]. Moreover, the composition of liposomes can be modeled to exert an immunostimulant effect (e.g., containing monophosphoryl lipid A), being useful as antigen carriers and adjuvants for vaccines [85]. For instance, a vaccine to prevent and reduce human immunodeficiency virus (HIV) spread has been designed (ACTHIVE-001). This vaccine consists in a native-like HIV-1 envelope adjuvanted with MPLA liposomes and is currently in phase I clinical trial (NCT03961438) to evaluate the safety and immunogenicity in healthy adults [72].

### 2.7. Liposomes for the Delivery of Immunosuppressive Drugs

Organ transplantation is considered one of the major advances of modern medicine and it is often the only chance for patient survival. Organ rejection still represents a challenge, and constant refinement of immunosuppression protocols is required [86] since treatment duration and intensity is associated with increased risk of developing malignancies [87]. Therefore, it is urgent to find tools that provide a more effective and safer use of immunosuppressant drugs. An example is the application of liposomes for the delivery for cyclosporin A. To prevent bronchiolitis obliterans syndrome after allogenic hematopoietic stem cell transplantation or after lung transplant, aerosolized liposomal cyclosporine A was developed (L-CsA) [88,89]. Phase II (NCT04107675) and phase III (NCT03657342, NCT03656926, NCT04039347) studies are being conducted to assess the tolerability and safety, as well as to evaluate the pharmacokinetics and therapeutic efficacy [72].

### 2.8. Liposomes for Diagnostic Applications

In Alzheimer’s disease, important diagnostic information can be obtained by different imaging modalities, such as magnetic resonance imaging (MRI) and positive emission tomography (PET), which allow the early detection of changes in the brain. One of the hallmarks of this irreversible neurodegenerative disease that impacts cognition and function is the progressive accumulation of extracellular amyloid beta plaques [90]. For contrast-enabled MRI of amyloid plaques, a novel liposomal platform loading the contrast agent gadolinium (ADx-001; single intravenous infusion) is under phase I clinical trial. This study will evaluate the safety and provide the proof-of-concept in patients with suspected Alzheimer’s disease (NCT05453539) [72].

Overall, liposomes have brought significant advances in medicine, with a positive outcome in terms of efficacy and safety. As previously mentioned, one of the benefits of liposomes is the ability to change the biodistribution profile of loaded drugs, depending on the lipid composition, leading to a higher concentration at target sites and minimizing exposure of healthy tissues in comparison with the unloaded drug. Nevertheless, this modified biodistribution may cause some unexpected effects [26]. This is the case of doxorubicin encapsulated in pegylated liposomes. Although its encapsulation in liposomes increases blood circulation and reduces the cardiotoxicity associated to this chemotherapeutic agent, a skin toxic effect has been described, named as hand-foot syndrome [26]. This effect is mainly due to the accumulation at hands and feet of the polymer PEG included in the lipid composition. Fortunately, this occurrence is frequently mild and patients are not usually required to withdraw treatment [26,91]. 

Another issue associated with liposome administration is the complement activation-related pseudoallergy (CARPA) [92]. This hypersensitivity reaction is sometimes observed upon first exposure to phospholipids included in liposome formulation. However, with subsequent administrations, the symptoms usually decrease or resolve [92,93]. In problematic clinical circumstances, CARPA can be managed by decreasing the infusion rate and by pre-treating patients with steroids and antihistamines to decrease its severity [92,93]. In preclinical studies, for the particular case of Doxil^®^, premedication with unloaded PEGylated liposomes (placebo liposomes) is being evaluated as a strategy to mitigate CARPA. These in vivo studies demonstrated that placebo liposomes induced tachyphylaxis, resulting in a substantial decrease or almost complete remission of the symptoms [94].

In summary, scientific knowledge is continuously evolving, providing solutions for current and emerging challenges in the area of nanomedicine. Preclinical research and clinical evaluation are crucial to further improve the efficacy and safety of liposomal formulations, ensuring their successful approval for the benefit of patients. All the limitations that may be observed for the new nanomedicines have to be evaluated case by case. Nevertheless, for each clinical condition, a rigorous balance between efficacy and safety have to be carefully considered.

## 3. Conclusions

Nanotechnology is, undoubtedly, vital to tackle complex medical situations. Liposomes, in particular, have revolutionized the pharmaceutical industry and medicine, providing innovative solutions for disease management and improving patients’ quality of life. Over the years, several liposomal products have successfully reached the market worldwide, with many being researched at the preclinical stage or undergoing clinical trials. These biocompatible lipid-based nanosystems improve the solubility and stability of drugs, prolong half-lives, and promote drug accumulation at target sites (e.g., tumor, infection, inflammation), enhancing treatment efficacy while minimizing unwanted side effects. From a regulatory point of view, the diversity of liposomal products requires appropriate guidelines to ensure their quality, effectiveness and safety, with each phase of research, development and production being tightly controlled. For instance, minor changes in the lipid composition might result in significant variations in the pharmacodynamics and pharmacokinetics of the loaded drug, having repercussions in the therapeutic performance. Globally, lipid-based nanosystems continue to provide innovative solutions for clinical challenges, advancing the treatment and diagnosis of human pathologies.

## Figures and Tables

**Figure 2 biomedicines-11-00435-f002:**
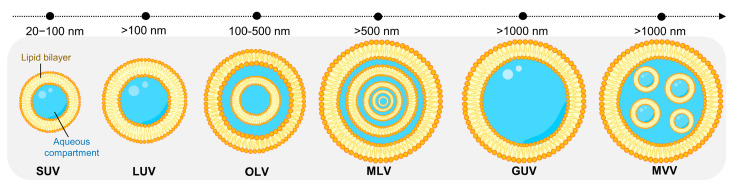
Visual representation of the different classes of liposomes based on size and lamellarity. SUV: small unilamellar vesicle; LUV: large unilamellar vesicle; OLV: oligolamellar vesicle; MLV: multilamellar vesicle; GUV: giant unilamellar vesicle; MVV: multivesicular vesicle.

**Figure 3 biomedicines-11-00435-f003:**
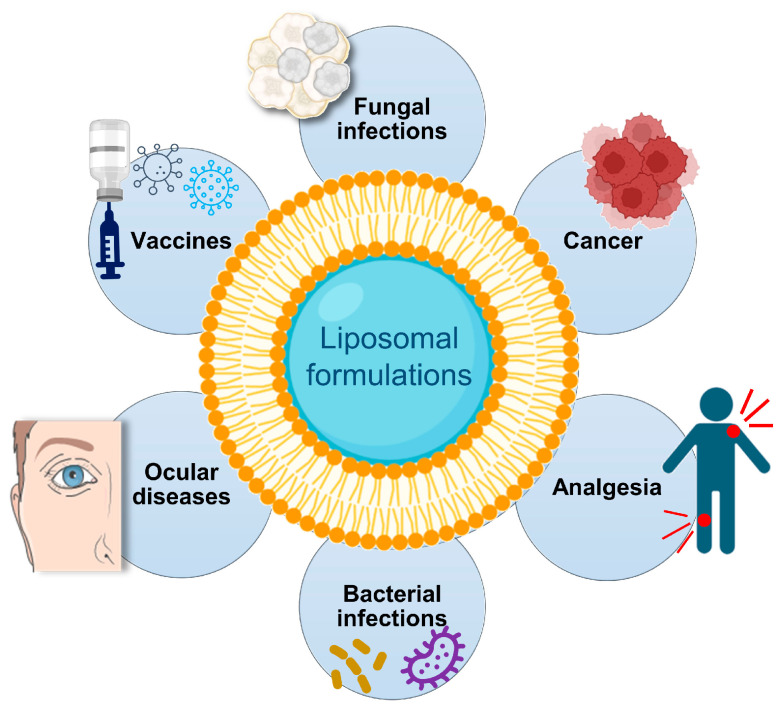
Clinical applications of liposomes.

**Table 1 biomedicines-11-00435-t001:** Classification of liposomes based on their composition and properties [22,26,27].

Classification of Liposomes	Composition and Main Properties
Conventional	Neutral or negatively charged phospholipids and/or cholesterol.
Long circulating	Surface is coated with inert, biocompatible polymers; displays dose-independent, non-saturable, log-linear kinetics and increased bioavailability.
pH−sensitive	Include phospholipids such as dioleoyl phosphatidyl ethanolamine (DOPE) with cholesteryl hemisuccinate (CHEMS) or oleic acid (OA); stable at neutral pH.
Cationic	Composed of cationic lipids appropriate for loading negatively charged macromolecules, such as DNA, RNA, and oligonucleotides.
Immunoliposomes	Conventional or long circulating liposomes with their surface coated with specific ligands (e.g., antibodies); can be recognized by receptors overexpressed at affected sites.

**Table 2 biomedicines-11-00435-t002:** Description of some applications of liposomes [27,35].

Applications	General Features
Drug Delivery	Encapsulation of drugs for in vivo delivery, a major use of these lipid-based nanosystems. Liposomes delay drug clearance, change its biodistribution profile, and minimize potential toxic effects, ultimately enhancing the therapeutic index [36].
Vaccines	Liposomes can carry and deliver antigens to antigen-presenting cells, whilst protecting them from degradation. Liposomes that are injected intramuscularly or subcutaneously accumulate in the lymph nodes, which is advantageous for vaccines [11].
Gene Therapy	Liposomes can be used as carriers for DNA and nucleic acid-based therapeutics, including anti-sense oligonucleotides and siRNA [11,19,37].
Diagnostic	Probes can be encapsulated for imaging applications, such as magnetic resonance imaging [38]
Supplements	Compounds such as glutathione and various vitamins have been associated to liposomes [39,40].
Cosmetics	Liposomes are being utilized to increase the topical delivery of main ingredients included in cosmetics [41].

**Table 3 biomedicines-11-00435-t003:** Selected preclinical studies of liposomes loaded with high and low molecular weight molecules.

Disease		Therapeutic Agent	Lipid Composition	Main Findings	Ref.
Cancer	Lymphoma	L-asparaginase	EPC:Chol:GM1 EPC:Chol:PI EPC:Chol:SA	Liposomal formulations increased up to 15-fold the half-life of the enzyme in blood circulation, enhanced its antitumor effect, and improved survival rate, with no adverse events.	[30]
		Acylated L-asparaginase	EPC:Chol:PI EPC:Chol:SA	Liposomal formulations increased up to 8-fold the half-life of the enzyme in blood circulation and enhanced the antitumor effect, with no adverse events.	[50]
	Lewis lung carcinoma	L-asparaginase	SPC:Chol:DSPE-PEG	Liposomes loading L-asparaginase increased survival rate.	[31]
	Melanoma	Hybrid molecule (L-tyrosine analogue conjugated with a triazene)	EPC:DSPE-PEG	Liposomes loaded with the hybrid molecule preferentially accumulated at tumor sites and significantly improved antimelanoma effect in subcutaneous and metastatic murine models devoid of toxicity.	[51]
		Copper(II) complex	DMPC:Chol:DSPE-PEG DMPC:CHEMS:DSPE-PEG	Liposomal formulations greatly enhanced the antimelanoma activity of metal complex with no adverse events.	[52]
		Iron(III) complex	DOPE:DOPC:CHEMS:DSPE-PEG	Liposomes loading the metal complex displayed the highest antitumor activity, even when compared with the positive control, TMZ. No toxic effects were reported.	[53]
	Colon cancer	Copper(II) complex	DMPC:DOPE:CHEMS:DSPE-PEG	pH-sensitive liposomes loaded with copper(II) complex impaired tumor progression compared to the compound in the free form without toxic effects.	[54]
		Zinc(II) complex	DOPE:DOPC:CHEMS:DSPE-PEG	pH-sensitive liposomes of zinc(II) complex reduced tumor progression in the same extent as the positive control 5-FU, using a 3-fold lower therapeutic dose and without toxic side effects.	[55]
Inflammation	Rheumatoid arthritis	Superoxide dismutase	EPC:Chol:SA EPC:Chol:PI	Liposomal formulations improved the therapeutic activity of the enzyme.	[42]
		Superoxide dismutase (SOD) or acylated superoxide dismutase (Ac-SOD)	EPC:Chol:DSPE-PEG EPC:Chol:SA	DSPE-PEG liposomes loading SOD and Ac-SOD displayed the highest half-life times in blood circulation. All SOD and Ac-SOD liposomes accumulated at inflammation sites. DSPE-PEG liposomes loading Ac-SOD showed a faster anti-inflammatory effect.	[33]
Infection	*Mycobacterium avium*	Paromomycin	DPPC:DPPG DMPC:DMPG:DSPE-PEG DPPC:DPPG:DSPE-PEG	Paromomycin-loaded liposomes significantly reduced bacterial loads in all infected organs, showing higher antimycobacterial activity than the positive control rifabutin.	[43]
		Rifabutin	PC:PS	RFB liposomal formulations reduced mycobacterial infection in a higher extent than the antibiotic in the free form both in therapeutic and prophylactic murine models.	[44]
	*Mycobacterium tuberculosis*	Rifabutin	DPPC:DPPG HPC:Chol:DSPE-PEG DPPC:PEG	DPPC:DPPG liposomes promoted a higher accumulation of RFB in liver, spleen and lung. This nanoformulation improved the antimycobacterial effect of RFB in the *M. tuberculosis* murine model.	[48]
	*Leishmania infantum*	Paromomycin	DPPC:DPPG	Paromomycin-loaded liposomes displayed a superior reduction of parasite burden, even when compared with the commercial antileishmanial drug Glucantime^®^.	[43]
Ischemia-reperfusion		Superoxide dismutase	EPC:Chol:DSPE-PEG (SOD liposomes) EPC:Chol:DSPE-PEG:DSPE-PEG-maleimide (SOD enzymosomes)	SOD enzymosomes enhanced the therapeutic effect of the enzyme, compared to SOD liposomes.	[45]
Thromboembolism		Streptokinase	DSPC:Chol:DSPE-PEG	Liposomes loaded with streptokinase increased 16-fold the half-life of the protein in blood circulation.	[46]
		Urokinase	DPPC:DSPE-PEG-NHS:DSPE-mPEG	Liposomal formulation improved the thrombolytic activity of urokinase, being safe.	[47]

EPC: egg phosphatidyl choline; Chol: cholesterol; GM1: monosialogangliosides; PI: phosphatidyl inositol; SA: stearylamine; SPC: soya phosphatidyl choline; DSPE-PEG: distearoyl phosphatidyl ethanolamine covalently linked to poly(ethylene glycol) 2000; DMPC: dimyristoyl phosphatidyl choline; CHEMS: cholesteryl hemisuccinate; HPC: hydrogenated phosphatidyl choline; SOD: superoxide dismutase; TMZ: temozolomide; 5-FU: 5-fluouracil; PS: phosphatidylserine; RFB: rifabutin; DSPE-PEG-NHS: distearoyl phosphatidyl ethanolamine covalently linked to succinimidyl poly(ethylene glycol) 2000; DSPE-mPEG: distearoyl phosphatidyl ethanolamine covalently linked to methoxy poly(ethylene glycol) 2000.

**Table 4 biomedicines-11-00435-t004:** Examples of liposomal formulations approved for clinical use [11,21,23,27,61,62].

Clinical Application	Trade Name	Active Pharmaceutical Ingredient	Lipid Composition	Year of First Approval
Cancer	DaunoXome^®^	Daunorubicin	DSPC:Chol	1996
	DepoCyt^®^	Cytarabine	DOPC:DPPG:Chol:triolein	1999
	Doxil^®^/Caelyx^®^	Doxorubicin	HSPC:Chol:DSPE-PEG2000	1995
	Myocet^®^		PC:Chol	2001
	Lipodox^®^		HSPC:Chol:DSPE-PEG2000	2012
	Lipusu^®^	Paclitaxel	PC:Chol	2006
	Mepact^®^	Mifamurtide	DOPS:POPC	2009
	Marqibo^®^	Vincristine	Sphingomyelin:Chol	2012
	Onivyde^®^	Irinotecan	DSPC:Chol:DSPE-PEG2000	2015
	Vyxeos^®^	Cytarabine and daunorubicin	DSPC:DSPG:Chol	2017
Fungal infections	AmBisome^®^	Amphotericin B	HSPC:Chol:DSPG	1990
	Amphocil^®^		Cholesteryl sulphate	1993
	Abelcet^®^		DMPC:DMPG	1995
	Amphotec^®^		Cholesteryl sulphate	1996
	Fungisome^®^		PC:Chol	2003
Bacterial infections	Arikayce^®^	Amikacin	DPPC:Chol	2018
Ocular disorders	Visudyne^®^	Verteporfin	DMPC:PG	2000
Analgesia	DepoDur™	Morphine sulfate	DOPC:DPPG:Chol:triolein	2004
	Exparel^®^	Bupivacaine	DEPC:DPPG:Chol:tricaprylin	2011
Vaccination	Epaxal^®^	Hepatitis A virus antigen, strain RGSB	DOPC:DOPE	1993
	Inflexal^®^ V	Influenza virus antigen, strains A and B	DOPC:DOPE	1997
	Mosquirix™	RTS,S antigen	DOPC:Chol	2015
	Shingrix	varicella zoster virus glycoprotein E	Chol:MPL:QS21	2017
	Comirnaty^®^	mRNA encoding for the SARS-CoV-2 spike protein	ALC-0315: ALC-015d:Chol:DSPC	2020
	Spikevax™		SM-102:PEG2000-DMG:Chol:DSPC	2021

PG: phosphatidyl glycerol; DSPC: distearoyl phosphatidyl choline; Chol: cholesterol; DOPC: dioleoyl phosphatidyl choline; DPPG: dipalmitoyl phosphatidyl glycerol; HSPC: hydrogenated soy phosphatidyl choline; DSPE-PEG2000: distearoyl phosphatidyl ethanolamine covalently linked to poly(ethylene glycol) 2000; PC: phosphatidyl choline; DSPG: distearoyl phosphatidyl glycerol; DMPC: dimyristoyl phosphatidyl choline; DMPG: dimyristoyl phosphatidyl glycerol; PG: phosphatidyl glycerol; DOPE: dioleoyl phosphatidyl ethanolamine; DEPC: 1,2-dierucoyl-sn-glycero-3-phosphocholine; DOPS: dioleoyl phosphatidyl serine; RTS,S: portion of *Plasmodium falciparum* circumsporozoite protein fused with hepatitis B surface antigen (RTS) and combined with hepatitis B surface antigen (S); MPL: monophosphoryl lipid A, a detoxified derivative of the lipopolysaccharide from *Salmonella minnesota*; QS21: saponin purified from the bark extract of *Quillaja saponaria* Molina (fraction 21); ALC-0315: 6-((2-hexyldecanoyl)oxy)-*N*-(6-((2-hexyldecanoyl)oxy)hexyl)-*N*-(4-hydroxybutyl)hexan-1-aminium; ALC-0159: 2 [(polyethylene glycol)-2000]-*N*,*N*-ditetradecylacetamide; SM-102: heptadecan-9-yl 8-{(2-hydroxyethyl)[6-oxo-6-(undecyloxy)hexyl]amino}octanoate; PEG2000-DMG: 1,2-dimyristoyl-rac-glycero-3-methoxypolyethylene glycol-2000.

## Data Availability

Not applicable.

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
