# Peer review of "Advancing Medicine with Lipid-Based Nanosystems—The Successful Case of Liposomes"

_biomedicines, 2023, doi:10.3390/biomedicines11020435_

Round 1

Reviewer 1 Report

The authors overview the development and application of liposomes in this manuscript. However, several issues need to be addressed.

1.     In section 2, the authors overview the advantage of several clinically applied liposomal drugs. It’s better to simultaneously describe the limitation or disadvantages of all those liposome drugs to help readers to understand liposomal medications.

2. Although there is no plagiarism, the description of historical liposome development and clinical application is similar to the preview review article.

Author Response

The authors overview the development and application of liposomes in this manuscript. However, several issues need to be addressed.

  1. In section 2, the authors overview the advantage of several clinically applied liposomal drugs. It’s better to simultaneously describe the limitation or disadvantages of all those liposome drugs to help readers to understand liposomal medications.

REPLY: We appreciate this pertinent suggestion. At the end of section 2. Liposomes as nanomedicine tools, we have included additional information about some issues related with liposomes in clinical, as well as solutions to manage these issues.

  1. Although there is no plagiarism, the description of historical liposome development and clinical application is similar to the preview review article.

REPLY: The authors appreciate this comment. Although a huge number of review publications focusing on liposomes and their applications as drug delivery systems against is published elsewhere, in the present Review article, we have tried to summarize the most important aspects of liposomes that turned into the successful lipid-based nanosystems that are in clinical use.

We expect that the reviewer understands our response.

Reviewer 2 Report

In this manuscript, Gaspar and co-workers did a nice work reviewing the recent advances in liposome-based nanomedicines. They first went through the history of liposome discovery, followed by a detailed review of liposome composition and classification. Next, they covered the recent liposome-based products in clinical use or undergoing clinical trials. Liposomes exhibit great biocompatibility and are capable of encapsulating a wide variety of cargo with various properties. Thus, they have been proven effective drug delivery carriers for cancers, gene therapies, vaccinations, and fungal/antibacterial treatments. The manuscript is well-written and easy to follow. The reviewer finds it can be a nice addition to the field and supports its publication in Biomedicines after some minor edits

1. In line 56, the authors mentioned that nonpolar chains have length variations and showed some examples. However, stearyl and oleyl tails have the same number of carbons but are different in saturation, which is another key difference for lipids and can significantly affect membrane properties (i.e., fluidity, etc.). The reviewer recommends the addition of some information on tail unsaturation. 

2. The DSPE-PEG2000 structure in Figure 1 has an amine terminal functional group. For general PEG lipids, the terminal group should be a methoxy instead of an amine. 

Author Response

In this manuscript, Gaspar and co-workers did a nice work reviewing the recent advances in liposome-based nanomedicines. They first went through the history of liposome discovery, followed by a detailed review of liposome composition and classification. Next, they covered the recent liposome-based products in clinical use or undergoing clinical trials. Liposomes exhibit great biocompatibility and are capable of encapsulating a wide variety of cargo with various properties. Thus, they have been proven effective drug delivery carriers for cancers, gene therapies, vaccinations, and fungal/antibacterial treatments. The manuscript is well-written and easy to follow. The reviewer finds it can be a nice addition to the field and supports its publication in Biomedicines after some minor edits.

  1. In line 56, the authors mentioned that nonpolar chains have length variations and showed some examples. However, stearyl and oleyl tails have the same number of carbons but are different in saturation, which is another key difference for lipids and can significantly affect membrane properties (i.e., fluidity, etc.). The reviewer recommends the addition of some information on tail unsaturation. 

REPLY: Thank you for mentioning this. Accordingly, we have added information regarding tail saturation. The following sentence was included:

“An important characteristic of lipids that affects the bilayer properties, including its fluidity, is the phase transition temperature (Tc). This parameter is defined as the temperature at which the physical state of the lipid changes from an ordered and rigid gel-state to a disordered and more fluid liquid-crystalline phase [14]. Tc highly depends on the length and saturation degree of nonpolar chains, with longer and more saturated chains corresponding to higher Tc [14]. Examples of phospholipids with different chain length and saturation degree are, distearoyl phosphatidyl choline (DSPC; Tc ≈ +55°C), dipalmitoyl phosphatidyl choline (DPPC; Tc ≈ +41°C), dimyristoyl phosphatidyl choline (DMPC; Tc ≈ +24°C), or dioleoyl phosphatidyl choline (DOPC; Tc ≈ −17 °C) (Figure 1b) [13,14].”

  1. The DSPE-PEG2000 structure in Figure 1 has an amine terminal functional group. For general PEG lipids, the terminal group should be a methoxy instead of an amine. 

REPLY: Thank you for this comment. We have replaced the DSPE-PEG structure in Figure 1.

Round 2

Reviewer 1 Report

All the suggestion has been added.